# Immune Profiling of COVID-19 in Correlation with SARS and MERS

**DOI:** 10.3390/v14010164

**Published:** 2022-01-17

**Authors:** Bariaa A. Khalil, Sarra B. Shakartalla, Swati Goel, Bushra Madkhana, Rabih Halwani, Azzam A. Maghazachi, Habiba AlSafar, Basem Al-Omari, Mohammad T. Al Bataineh

**Affiliations:** 1Sharjah Institute for Medical Research, College of Medicine, University of Sharjah, Sharjah P.O. Box 27272, United Arab Emirates; U19106038@sharjah.ac.ae (B.A.K.); u19106034@sharjah.ac.ae (S.B.S.); u19106040@sharjah.ac.ae (S.G.); u19106031@sharjah.ac.ae (B.M.); rhalwani@sharjah.ac.ae (R.H.); amagazachi@sharjah.ac.ae (A.A.M.); 2Faculty of Pharmacy, University of Gezira, Wad Medani 2667, Sudan; 3College of Medicine, University of Sharjah, Sharjah P.O. Box 27272, United Arab Emirates; 4College of Medicine and Health Sciences, Khalifa University, Abu Dhabi P.O. Box 127788, United Arab Emirates; Habiba.alsafar@ku.ac.ae or; 5Center for Biotechnology, Khalifa University of Science and Technology, Abu Dhabi P.O. Box 127788, United Arab Emirates; 6Emirates Bio-Research Center, Ministry of Interior, Abu Dhabi P.O. Box 389, United Arab Emirates; 7KU Research and Data Intelligence Support Center (RDISC) AW 8474000331, Khalifa University of Science and Technology, Abu Dhabi P.O. Box 127788, United Arab Emirates

**Keywords:** chemokines, cytokines, COVID-19, MERS-CoV, SARS-CoV-2, SARS-CoV

## Abstract

Acute respiratory distress syndrome (ARDS) is a major complication of the respiratory illness coronavirus disease 2019, with a death rate reaching up to 40%. The main underlying cause of ARDS is a cytokine storm that results in a dysregulated immune response. This review discusses the role of cytokines and chemokines in SARS-CoV-2 and its predecessors SARS-CoV and MERS-CoV, with particular emphasis on the elevated levels of inflammatory mediators that are shown to be correlated with disease severity. For this purpose, we reviewed and analyzed clinical studies, research articles, and reviews published on PubMed, EMBASE, and Web of Science. This review illustrates the role of the innate and adaptive immune responses in SARS, MERS, and COVID-19 and identifies the general cytokine and chemokine profile in each of the three infections, focusing on the most prominent inflammatory mediators primarily responsible for the COVID-19 pathogenesis. The current treatment protocols or medications in clinical trials were reviewed while focusing on those targeting cytokines and chemokines. Altogether, the identified cytokines and chemokines profiles in SARS-CoV, MERS-CoV, and SARS-CoV-2 provide important information to better understand SARS-CoV-2 pathogenesis and highlight the importance of using prominent inflammatory mediators as markers for disease diagnosis and management. Our findings recommend that the use of immunosuppression cocktails provided to patients should be closely monitored and continuously assessed to maintain the desirable effects of cytokines and chemokines needed to fight the SARS, MERS, and COVID-19. The current gap in evidence is the lack of large clinical trials to determine the optimal and effective dosage and timing for a therapeutic regimen.

## 1. Introduction

Coronaviruses (CoVs) are a group of enveloped, positive-sense, single-stranded RNA viruses belonging to *Coronavirinae*, a subfamily of *Coronaviridae* [1]. CoVs have the largest identified viral RNA genome with the ability to exert a variety of diseases with different severity in animals and humans [1,2]. CoVs are classified into α, β, γ, and δ coronaviruses, with β-CoVs further subdivided into A, B, C, and D lineages [2]. Mainly α and β infect humans with six genera identified; two of them belong to α-CoVs (HCoV-229E and NL63) and the other four are part of the β-CoVs (HCoV-OC43, HCoV-HKU1, MERS-CoV, and SARS-CoV) [1,3,4].

SARS-CoV and MERS-CoV cause severe acute respiratory syndrome (SARS) and the Middle East respiratory syndrome (MERS) respectively via zoonotic transmission [4,5,6,7]. The two viruses are mainly transmitted by respiratory droplets and possibly through feces among patients who present with varying symptoms ranging from a mild flu-like illness to atypical pneumonia, which can progress to acute respiratory distress syndrome (ARDS), multi-organ failure, and death [4,7,8]. The SARS outbreak which emerged in China in 2002 reported 8098 infected cases and a cumulative fatality rate of 9.6%, whereas the MERS epidemic in 2012 reported 2494 confirmed cases and a fatality rate of 37.1%, with the majority of cases being in Saudi Arabia [9,10].

In December 2019, a new β-CoV, the Severe Acute Respiratory Syndrome Corona Virus 2 (SARS-CoV-2), emerged for the first time in Wuhan, China, and then was announced as a pandemic in March 2020 [11]. More than two years into this pandemic, healthcare systems across the world continue to be overwhelmed with soaring daily cases [12]. Similar to SARS and MERS, the first case of SARS-CoV-2 occurred via zoonotic transmission linked to a seafood market and was capable of human-to-human transmission mainly via respiratory droplets [13,14]. SARS-CoV-2 causes the syndrome coronavirus disease 2019 (COVID-19), as named by the WHO [15]. Similar to SARS-CoV and MERS-CoV, SARS-CoV-2 is considered a highly pathogenic human corona virus that shares 79.5% of the SARS-CoV genetic sequence and 50% with that of MERS-CoV [16]. However, it is more aggressively transmitted, as over 267 million cases were confirmed as having caused over 5.2 million deaths in over 216 countries by the 10th of December 2021 [17]. The fatality rate in males is 2.4 times higher than in females [18], and there are higher co-morbidities in people over 60 years of age [13,19].

The median incubation period post-SARS-CoV-2 infection is around 4–5 days, the symptoms appear within 11.5 days [20,21,22,23], the viral load reaches its peak in 5–6 days following the onset of symptoms as compared to 10 days for SARS-CoV [24,25,26,27], and ARDS occurs 8–9 days after symptoms’ onset in severely diseased cases [28,29]. Generally, a majority of COVID-19 patients are asymptomatic or mildly symptomatic, with or without flu-like symptoms (81.4%) [30]. The virus mainly infects the upper respiratory tract, but in 13.9% of patients, the infection can extend to the lower respiratory tract, causing severe pneumonia and leading, in some cases, to fatal acute lung injury (ALI) and ARDS, where patients require assisted ventilation and intensive-care therapy [29,30]. ALI and ARDS can progress to multi-organ failure or disseminated intravascular coagulation (DIC) with rare clinical recovery, making SARS-CoV-2 a major public health threat [20,31,32].

The main underlying cause of ARDS in SARS, MERS, and COVID-19 is the cytokine storm caused by proinflammatory mediators that exacerbate the host immune response [29,33,34,35]. Generally, cytokines and chemokines play an important role in the immunopathology of diseases caused by viral pathogens. Cytokines are molecular messengers, including interferon (IFN), interleukin, and growth factors, used by immune cells to communicate in a paracrine or autocrine fashion [36]. On the other hand, chemokines are small proteins that bind G-protein-coupled receptors (GPCR) to stimulate cell migration [37,38]. Chemokines share four cysteines with two characteristic disulfide bonds important for the conserved chemokine fold, and the spacing of the first two cysteines (adjacent (CCL), separated by an amino acid (CXCL) or separated by three amino acids (CX3CL)) is the basis for their systematic nomenclature [39].

The aim of this review is to examine the role, differences, and similarities in the profile of cytokines and chemokines during the pathogenesis of SARS, MERS, and COVID-19. It further highlights the most important findings to date about COVID-19 pathogenesis to better understand the virus behavior and identify therapeutic targets. This review discusses, in detail, the main treatment options targeting the chemokines and cytokines implicated in COVID-19 and their mechanism of action in ameliorating the severity of the disease; it then discusses the limitations and challenges in the current literature.

## 2. Search Strategy

In this narrative review, a comprehensive literature search was conducted. Following Gasparyan and colleagues’ recommendations [40], PubMed (MEDLINE), EMBASE, and Web of Science were electronically searched, without language or date restrictions. The keywords related to “Chemokines”, “Cytokines” and “COVID-19”, “MERS-CoV”, “SARS-CoV-2”, and “SARS-CoV” were used with Boolean combinations. Additionally, several authors of this review are experts in the field, and opinions expressed in this review are also based on personal experience of writing, editing, and commenting on review articles.

## 3. The Innate and Adaptive Immune Responses Associated with SARS-CoV-2 and the Correlation with SARS-CoV-2 and MERS-CoV Infections

The entry of the SARS, MERS, and COVID-19 β-CoVs is facilitated by the S protein, which enables their replication inside host cells following the release of the viral RNA [41]. The S1 subunit of the homo-trimeric S protein of SARS-CoV binds to angiotensin-converting enzyme 2 (ACE2) receptors on the alveolar cells, and the S2 subunit assists in the fusion of the viral membrane with the host cell [42,43]. Similarly, SARS-CoV-2 uses the S protein to bind ACE2, yet attaches to the host cell with a higher affinity compared to SARS-CoV [44,45,46,47]. ACE2 is expressed on a variety of cells in the lungs and the gut, including epithelial cells of the airways and alveoli, vascular endothelial cells, and macrophages [48,49,50,51]. The transmembrane serine protease 2 (TMPRSS2) is also needed for SARS-CoV-2 entry [52]. On the other hand, the S1 subunit of the S protein of MERS-CoV binds to dipeptidyl peptidase 4 (DPP4/CD26), a type-II transmembrane glycoprotein that is expressed on multiple cells, including macrophages, fibroblasts, epithelial cells, and endothelial cells [4,53,54]. Moreover, MERS-CoV and SARS-CoV-2 exhibit the furin-like cleavage site, which is absent in SARS-CoV. This explains the difference in cellular tropisim and pathogenesis, since the furin-like cleavage site enhances viral fusion with host cell membrane [55].

Upon entry into the host’s lung cells, the three β-CoVs cause cell destruction and trigger a local immune response. Despite the activation of lung macrophages by the S protein of SARS-CoV, phagocytosis is dampened with the increase in cytokine production [56,57]. Similarly, the phagocytic function of macrophages and their ability to produce tumor necrosis factor-alpha (TNF-α) and interleukin 6 (IL-6) were suppressed upon MERS-CoV binding to DPP4, indicating that MERS-CoV S protein could induce immune suppression by initiating signaling through DPP4 [54]. In addition, IL-1R-associated kinase (IRAK-M) and peroxisome proliferator-activated receptor-γ (PPARγ), which is known to inhibit the activation of macrophages by toll-like receptor (TLR), were found to be upregulated by the S protein, suggesting another molecular mechanism by which the immune system and cytokine production can be suppressed by MERS-CoV [54]. In SARS-CoV-2, the spleen and lymph nodes of six patients who died from COVID-19 showed a cluster of differentiation (CD)68^+^ and CD169+ macrophages expressing ACE2-SARS-CoV-2 complex, reflecting their importance in the viral spread and inflammation [58]. Furthermore, in comparison to mildly infected patients, severe COVID-19 patients showed an increase in the populations of the highly inflammatory monocyte-derived FCN1+ macrophages in their Broncho Alveolar Lavage Fluid (BALF) and the CD14^+^CD16^+^ inflammatory monocytes in their peripheral blood [59]. Dendritic cells (DCs) also play a crucial role in viral spread and replication. Upon entry of SARS-CoV, DCs undergo functional modification, using DC-specific intracellular adhesion molecule-grabbing non-integrin CD209 receptors [60,61]. Furthermore, productive replication of MERS-CoV was demonstrated in monocyte-derived DCs [62]. In comparison with SARS-CoV, higher surface expression of major histocompatibility complex (MHC) class II Human Leukocyte Antigen—DR isotype (HLA-DR) and the co-stimulatory molecule CD86 were reported upon MERS-CoV infection, suggesting that the virus has a greater ability to activate monocyte-derived-DCs [62].

The activation of the innate immune cells induces the release of cytokines and chemokines that bring the adaptive immune system into action to fight viral infection. In SARS-CoV, two epitopes identified in the S protein resulted in high IFN-γ production and T-cell response, which further augmented the humoral immunity against SARS-CoV infection [63]. However, the frequency of CD4^+^ and CD8^+^ T cells is reduced and priming by DC is also impaired [64]. For an unknown reason, the number of natural killer (NK) cells responsible for eliminating virus-infected cells is decreased in SARS-CoV [65]. The role of the adaptive immune system in MERS-CoV infection and, specifically, the role of cytotoxic CD8^+^ T cells, which are mainly responsible for viral clearance, were demonstrated in a replication-deficient adenovirus (Ad5-hDPP4)-transduced mouse model deficient for T and B cells [66]. Further, a transcriptomic study on bronchial epithelial cells infected with MERS-CoV indicated that the virus downregulates antigen-presenting proteins and major histocompatibility complex (MHC) I/II, and this, in turn, inhibits T-cell responses and allows the virus to evade the immune system [67]. Moreover, MERS-CoV impairs the function of CD3^+^ T cells upon binding to DPP4 and downregulating it, particularly because DPP4 has a critical role in the signal transduction pathways involved in T-cell activation [68,69]. Not only does MERS-CoV impair the function of CD3^+^ T cells, but it also induces intrinsic and extrinsic apoptosis, resulting in lymphopenia [70]. Post-SARS-CoV-2 infection, T- and B-cell responses are detected in the blood of infected patients around one week after the onset of symptoms [71]. Cytokine-secreting immune cells, such as CXCR3^+^CD4^+^ T cells, CXCR3^+^CD8^+^ T cells, and CXCR3^+^ NK cells, were shown to be elevated in severe COVID-19 patients [72]. However, lymphocytes required to clear the virus, including CD4^+^ and CD8^+^ T cells, NKs, and B cells, were decreased in a directly proportional manner to disease. Collectively, these strategies delay the response of the adaptive immune system, resulting in inefficient viral clearance and the dissemination of the infection to extra pulmonary sites.

## 4. The Role of Cytokines and Chemokines in SARS

Studies on mice and patients post-SARS-CoV infection showed that proinflammatory cytokines and chemokines produced by airway epithelial cells (AECs), DCs, and macrophages have an important role in lung immunopathology and disease severity. IL-6, IL-8, IL-1β, and TNF-α released by epithelial cells, pneumocytes, and macrophages of the lung and bronchial tissue are among the initial cytokines rapidly increasing in the blood of patients during early infection with SARS-CoV [73,74,75,76]. IL-6 and IL-1β stimulate the production of C-reactive protein (CRP), which mediates systemic inflammatory responses, while TNF-α stimulates fibroblast proliferation and collagen fiber synthesis, which subsequently cause pulmonary fibrosis [27,77,78,79]. As a result of the modification in their functions as discussed above, DCs release chemokines, such as CCL3 (macrophage inflammatory protein (MIP-1α)), CCL5 (RANTES), CXCL10 (IP-10), and CCL2 (Monocyte Chemoattractant Protein-1 (MCP-1)), which induce the migration of inflammatory leukocyte cells [61]. By autocrine manner, these chemokines also enhance DC migration to lymph nodes to prime and activate T cells [80]. CCL2, CCL3, CCL5, CXCL10, and CCL10 are among the chemokines that drastically increased within 24 h and remained elevated after 48 h of infection [81]. Collectively, the increase in the aforementioned inflammatory mediators is an indicator of the host antiviral response.

In comparison to uncomplicated cases, severely infected patients with ARDS exhibit higher levels of the cytokines, i.e., IFN-α, IFN-γ, IL-1, IL-6, IL-12, and TGFβ, and the chemokines, i.e., CCL2, CXCL10, CXCL9, and IL-8 [34,82,83,84,85,86]. High concentrations of CCL2 and TGF-β1 released by SARS-CoV-infected ACE2-expressing cells induce the migration of monocytes and macrophages from the blood stream to the injured lung, where they proliferate (Mac387^+^) and get activated (CD25^+^) to produce additional proinflammatory mediators that can worsen the disease [87,88,89]. In addition to its role in recruiting and activating macrophages and monocytes, TGF-β1 contributes to the lymphopenia and thrombocytopenia detected in SARS patients by enhancing Fas-mediated cell apoptosis, consequently leading to the death of alveolar epithelial cells, lymphocytes, and platelets [90,91,92]. Another possible underlying cause for lymphopenia is the impaired DC function, being the only antigen-presenting cell capable of priming T cells, and this is probably attributed to the action of proinflammatory cytokines IL-6 and IL-8 [93]. Further, the cytokines produced by Th2 cells, such as IL-4 and IL-10, were shown to decrease post-SARS infection, suggesting that the immune response is dominated by Th1 rather than Th2 cells [83,94]. Moreover, IL-4 and with IFN-γ were demonstrated to inhibit SARS-CoV replication partly by ACE2 downregulation [95]. However, elevated levels of IL-10 were also detected in some SARS patients, and this was attributed to the dual effect of IL-10 on T lymphocytes, whereby it inhibits the production of IL-2, IFN, and TNF from Th1 and activates cytotoxic CD8^+^ T cells and NKs, hence increasing susceptibility to the disease [94]. The discrepancy in IL-10 fluctuation post-SARS infection is also seen in IL-2 expression, where some studies reported high-expression post-SARS onset, whereas others did not [34,96,97].

Regarding IFNs, elevation in the levels of IFN-α2, IFN-β1, and IFN2 was demonstrated within 24 h of infection suggesting the involvement in plasmacytoid DCs and monocytes in the early stage of the disease [81]. Furthermore, the binding of Toll-like receptor 3 (TLR3), TLR7, and retinoic acid–inducible gene I (RIG-1)-like receptors (RLRs) to SARS-CoV-pathogen-associated molecular patterns (PAMPS) and viral RNA initiates a downstream signaling cascade, resulting in the production of proinflammatory cytokines, mainly type 1 interferon (IFN), which limits the active replication of the virus [98,99]. However, post-SARS-CoV infection, IFN secretion is dysregulated, and excessive proinflammatory cytokines are released, leading to inefficient innate immune response [100]. Dysregulation in IFN production (α and γ) and IFN-stimulated genes (ISGs) transcription is mediated through different mechanisms by SARS-CoV and have a major contribution to disease pathogenesis [76,98]. The binding of SARS-CoV RNA to TLR3, RIG-I, and MDA-5 receptors leads to the phosphorylation, activation, and nucleus translocation of the IFN regulatory factor 3 (IRF-3) and IRF7, resulting in IFN-α/β synthesis [101,102]. IFN-α/β binds to IFN receptors (IFNR) present on all nucleated cells and activates signal transducer and activator of transcription (STAT) proteins through phosphorylation by Janus kinase 1 (JAK1), thus promoting the transcription of IFN-stimulated genes (ISGs) that exhibit antiviral properties [103]. However, SARS-CoV forms perinuclear double-membrane vesicles (DMVs) in the host cell for its RNA synthesis [104]. This may help in avoiding recognition by PPRs and the production of IFN-α/β [105]. Moreover, the open reading frame (ORF) 3b and ORF6 encoding the nucleocapsid (N) protein of SARS-CoV inhibit IRF-3 translocation to the nucleus and downregulate the production of TGF-β [106]. Furthermore, ORF6 inhibits STAT1 translocation to the nucleus and suppresses ISGs transcription [107]. Another mechanism by which SARS-CoV inhibits IFN secretion is via its membrane papain-like protease (PLpro-TM) that inhibits the STING/TBK1/IKKε pathway required for the phosphorylation and dimerization of IRF3 [108]. Studies on mice infected with SARS also showed that virus replication is associated with delayed IFN1 signaling, which is responsible for the accumulation of inflammatory monocytes and macrophages, elevation in the levels of cytokines and chemokines, impairment of adaptive immunity, and development of vascular leakage [109]. IFN also tends to upregulate inhibitory molecules (PDL-1) on T cells, resulting in impaired adaptive immune response upon viral infection [110]. Moreover, interferon-γ-induced protein CXCL10 secreted by monocytes, endothelial cells, and fibroblasts is prominent in SARS-CoV patients and one of the early chemokines increased in blood and lung tissue with its level increasing with the rise in infection and returning to normal during recovery [73,111]. Upon binding to its receptor CXCR3, CXCL10 recruits monocytes, macrophages, DCs, NK cells, and T lymphocytes toward interstitial lung tissue and induces inflammation in SARS patients [112].

## 5. The Role of Cytokines and Chemokines in MERS

Upon infecting the human airway epithelial cells, MERS-CoV induces delayed, yet significant and higher levels of the proinflammatory cytokines IL-1β, IL-8, and IL-6 as compared to SARS-CoV [113]. IL-1β and IL-8 are important mediators of ARDS, whereby IL-8 recruits and activates neutrophils, which in turn recruit more immune cells [6,113,114,115,116]. As for IL-6, a significant release by MERS-infected human-monocyte-derived macrophages and by the lung lesions of infected animals was detected [117,118]. IL-6 is associated with disease progression and severity, since higher levels were observed in the second and third week of illness in a severe MERS group of patients compared to the mild group [119]. Furthermore, very low levels of innate antiviral cytokines, such as TNF-α, IFN-β, and CXCL10, were induced by MERS-CoV compared to SARS-CoV [113]. In line with these findings, MERS-CoV was unable to induce antiviral TNF-α, IFN-β, and CXCL10 in the human alveolar basal epithelial cell line (A549 cell line) [120]. These findings indicate that MERS attenuate innate immunity and has a greater ability to evade the antiviral response compared to SARS-CoV. However, CXCL10 was also reported to be associated with disease severity, since persistent high levels were reported in the serum of a patient who died post-MERS infection [121]. This was further supported by a serum analysis study on nine severely infected MERS patients with higher levels of CXCL10 observed as the disease progress specifically in the second and third week after onset of symptoms [119]. Another in vitro study comparing the levels of cytokines and chemokines released by blood-monocyte-derived macrophages and DCs upon MERS-CoV or SARS-CoV infection demonstrated higher levels of IFN-γ, IL-12, IL-8, CCL2, CCL3, CCL5, and CXCL10 post-MERS-CoV as compared to SARS-CoV, while comparable levels of TNF-α and IL-6 were displayed [62,117]. Moreover, IFN-α, CXCL10, IL-6, IL-8, and CCL2 were detected on day 11 in patients’ serum, and their levels decreased upon clinical improvement [122]. Cytometric bead array analysis of the cytokine profile of seven laboratory-confirmed MERS patients further showed high levels of the cytokines IFN-α2, IFN-γ, IL-10, TNF-α, IL-15, and IL-17, whereas no difference in the levels of IL-12, IL-2, IL-4, IL-5, IL-13, and TGF-α was observed [33]. IL-17 plays an important role in recruiting neutrophils and monocytes, which produce IL-1β, IL-6, TNF-α, TGF-β, IL-8, and CCL2 collectively implicated in airway remodeling [33,123]. Clinically, high levels of cytokines and chemokines in the serum of MERS-CoV-infected patients were found to be associated with increased neutrophil and monocyte counts in the peripheral blood and lungs of MERS patients [113,119,124]. It is worth noting that the discrepancy in the levels of some chemokines and cytokines, such as CXCL10, TNF-α, and IL-6, between in vitro and in vivo studies could be attributed to the absence of the physiological setting or different measurement timing. This suggests that there is an interplay between the different inflammatory mediators, and studying a particular chemokine or cytokine at a time will probably provide an incomplete conclusion about its role in disease pathogenesis.

Regarding IFN, an in vitro study on monocyte-derived-DCs infected with MERS-CoV showed no expression of the anti-inflammatory IFN-β, marginal expression in IFN-α, and higher expression in IFN-γ as compared to SARS-CoV [62]. Another study assessing the levels of IFN-α in two MERS patients showed that a higher expression level of IFN-α in the broncho-alveolar lavage cells was correlated with survival of one patient, whereas its absence was associated with the death of the other [121]. The reduction in IFN-α expression was explained by the decrease in IRF-3 and IFR-7 observed in the deceased patient [121]. Furthermore, IL-12 and IFN-γ were also detected with IFN-α, suggesting a critical role of IFN-α in the development of an early antiviral Th1 adaptive immune response mediated by the release of IL-12 and IFN-γ against MERS-CoV infection [121]. An elevation in the level of IL-10 was detected in the serum of a MERS-CoV patient 0–3 days post-infection and was found to be correlated with persistence of viral infection, especially since higher levels were observed in the patient who did not tolerate the infection compared to the recovered patient [121]. Not only does IL-10 activate the JAK/STAT signaling pathway, resulting in the production of more proinflammatory cytokines, but it can also inhibit IFN-γ production thereby reducing CD8^+^ T-cell proliferation and increasing MERS-CoV replication [121,125]. Taken together, it can be suggested that IFN-α is a major component of the antiviral host response against MERS-CoV, as its absence is associated with a worse outcome.

## 6. The Role of Cytokines and Chemokines in COVID-19

Pyroptosis or inflammatory programmed necrosis is an important process that contributes to the inflammation associated with SARS-CoV-2 infection and to the activation of the adaptive immune system [126,127]. The destruction of virally infected cells produces PAMPs, viral RNA, and damage-associated molecular patterns (DAMPs) which bind to their respective pattern-recognition receptors (PRRs) on alveolar epithelial cells and macrophages and induce pyroptosis and the release of proinflammatory cytokines, mainly IL-1β and IL-18 [29]. Furthermore, other cytokines and chemokines, such as IL-6, IFN-γ, CCL2, and CXCL10, are released and polarize T helper cells to activate B cells and cytotoxic CD8^+^ T cells, thus bringing the adaptive immune system into action in the airways to control the viral infection [128]. Real-time polymerase chain reaction (RT-PCR) and next-generation sequencing on SARS-CoV-2-infected patients showed an initial elevation in the plasma levels of IL-1β, IL-1RA, IL-7, IL-8, IL-10, IFN-γ, CCL2, CCL3, CCL4, CCL7, and granulocyte colony-stimulating factor (G-CSF), whereas the levels of IL-6, IL-2, IL-7, IL-17, IL-10, CCL3, IL-8, CXCL10, and TNF-α were shown to be continuously elevated in severely infected patients [29]. The elevation of these cytokines and chemokines indicates hyperactivation of Th1 cells [129]. Transcriptomic analysis on patients’ BALF also showed elevated levels of TGF-β, CXCL1, IL-10, CCL2, CXCL2, CCL8, CCL3, CCL4, CXCL10, IL-33, IL-8, and CXCL6 [130]. Moreover, CXCL17, a macrophage chemoattractant, is among the chemokines that were firstly upregulated in all SARS-CoV-2 patients whose BALFs were analyzed for meta-transcriptome sequencing, and functional analysis suggests an important role of CXCL17 in COVID-19 pathogenesis [131]. IL-8, CXCL1, CXCL2, and their respective receptors were also upregulated in the same patients, being critical for the recruitment of neutrophils to the lungs [131,132,133,134]. This finding was consistent with the high neutrophil-to-lymphocyte ratio particularly in patients with high viral load, cytokine levels, and ISG expression [131]. In addition to CXCL8, IL-10, TNF-α, IL-15, and IL-27 were shown to be positively correlated people of an older age (above 60) who present a significant reduction in the total T-lymphocyte number and an increased expression of T-cell exhaustion markers as compared to younger infected individuals [135]. Defining age-associated immune profile in SARS-CoV-2-infected patients aids in identifying preventive and therapeutic strategies, especially that age is a key factor in COVID-19 morbidity and mortality. Collectively, all the cytokines and chemokines contribute to COVID-19 immunopathology, yet IL-6, IFN, IL-17, TGF-β, TNF-α, and CXCL10 are believed to have major roles in the lung pathogenesis post-SARS-CoV-2 infection.

### 6.1. IL-6

COVID-19 activates CD4^+^ T cells to differentiate into pathogenic Th1 cells, which release GM-CSF and other proinflammatory cytokines that further activate monocytes to release IL-6 [136]. IL-6 can also be released from macrophages and DCs infected with SARS-CoV-2 [137]. IL-6 binds to its receptors on immune and non-immune cells and activates the downstream JAK-STAT3 and JAK-SHP-2 mitogen-activated protein (MAP) kinase pathway, resulting in the release of vascular endothelial growth factor (VEGF), CCL2, IL-8, and additional IL-6 [138]. Furthermore, IL-6 also decreases the expression of E-cadherin on endothelial cells, leading, together with VEGF, to an increase in vascular permeability and leakage, eventually resulting in the hypotension and pulmonary dysfunction seen in SARS-CoV-2 infection [138]. A meta-analysis of nine studies from China that studied 1426 patients supported the role of IL-6 in COVID-19 virulence. It was demonstrated that IL-6 was elevated in the serum of severely infected patients experiencing respiratory distress and admitted to the ICU, hence making IL-6 an important marker to evaluate disease severity and early stratify patients at risk to progress into complications [139]. Moreover, IL-6 is responsible for the elevation in CRP, serum amyloid A, fibrinogen, and hepcidin and the inhibition of albumin synthesis [139]. Conversely, another study that investigated the link between lymphocyte subsets, cytokine release, pulmonary inflammation index (PII), and disease evolution showed a zero value of IL-6 in mildly infected patients in half of the study population. They determined this finding to be caused by the inhibition of Th2 involved in humoral immunity in an early stage of infection, hence emphasizing the importance of IL-6 as a marker of disease severity [140]. In addition, no significant change in the transcription levels of IL-6 in PBMC of COVID-19 patients was detected, indicating that the source of IL-6 in the serum is mainly the lung epithelial cells [130]. Furthermore, the expression level of IL-6R was lower in the BALF of COVID-19 patients compared to controls with no difference on PBMCs, suggesting that the IL-6/IL6R axis in the epithelial cells of the lungs is the one involved in the immunopathology of the disease [130].

Another effect of IL-6 is the ability to inhibit HLA-DR expression on CD14 monocytes [141]. This was detected in COVID-19 patients with high serum levels of IL-6 and low HLA-DR on their CD14 monocytes, along with a low lymphocyte count [142]. In contrast, less severe patients with lower levels of IL-6 showed higher circulating HLA-DR cells [71]. Not only does IL-6 cause defective lymphoid function, but it also impairs the function of NK cells in clearing virally infected cells and reduces their cytotoxic effect specifically by inhibiting the perforin and granzyme B through blocking the STAT5 signal transduction required for regulating perforin transcription [143,144]. These results pinpoint the role of IL-6 not only in inducing the cytokine storm in COVID-19 patients but also in affecting the function of lymphoid cells, suggesting that IL-6 plays a key role in dysregulating adaptive immunity and consequently dampening the host’s ability to fight the virus.

### 6.2. IFN

Type 1 and type III IFN responses are considered to be the major first antiviral defense mechanism elicited by the innate immune sensors [145]. Upon binding to the ubiquitously expressed type I IFN receptor (IFNAR), type I IFNs (IFN-α, IFN-β, IFN-ε, IFN-κ, and IFN-ω in humans) activate interferon-stimulated genes (ISGs), which interfere in viral replication [146]. On the other hand, type III IFNs (IFN-λ) bind to the type III IFN receptor (IFNLR) expressed on epithelial cells and certain myeloid cells [147]. Accordingly, suppressing or modulating the functions of IFNs and ISGs is considered one successful mechanism for viral pathogens to bypass the immune system. This applies as well to the highly pathogenic coronaviruses, which use various mechanisms to evade and suppress the IFN response, including IFN production, signaling, and ISG effector function, despite the powerful host antiviral strategy [145]. However, in addition to viral factors, host factors are important determinants of the IFN signaling outcome in being protective or pathogenic against a particular viral pathogen. For instance, the age of the host affects the cytokine profile and disease pathogenesis, due to differences in the imbalance between pro-inflammatory versus IFN response in different age groups. A study on aged macaques that were infected with SARS-CoV showed more lung pathology and higher expression of pro-inflammatory cytokines but lower expression of IFN-Is in older macaques compared to younger macaques [148].

Several studies highlighted the role of IFN in the immunopathology of COVID-19 patients. During SARS-CoV-2, interferon signaling was detected to be the highest upregulated pathway by the global functional analysis performed on meta-transcriptome sequencing data on the BALF of COVID-19 patients [131]. This observation was accompanied by a marked elevation in ISGs and interferon-induced transmembrane proteins (IFITMs), which were shown to inhibit the cellular entry of SARS-CoV and MERS-CoV [131,149,150]. It is worth mentioning that the IFN response triggered in SARS-CoV-2 is more robust and protective as compared to SARS-CoV; this might explain the lower proportion of severe cases and fatality rate of COVID-19 compared to SARS [32]. Nevertheless, the study did not detect significant upregulation of IFNs, despite the increase in ISGs, suggesting that SARS-CoV-2 might be delaying IFN production by inhibiting innate immune signaling, and this discrepancy needs further investigation [131]. Another study revealed upon conducting transcriptome profiling of various cell types that SARS-CoV-2 infection elicits very low IFN-I or IFN-III and limited ISG response while inducing chemokine and pro-inflammatory cytokine genes [151]. Interestingly, a small COVID-19 patient cohort revealed that levels of IFN-α and ISGs were associated with the viral load, as well as disease severity, thus indicating that severe infections lead to high IFN signatures but fail to bring down viral load [152].

Regarding the possible mechanism that SARS-CoV-2 uses to antagonize the antiviral effect of IFN, it was postulated that the non-structural proteins 4a (NS4a) and NS4b shared between MERS and SARS-CoV-2 that are known to suppress IFN-λ production from lung epithelial cells are being used by SARS-CoV-2 to evade the immune system [153,154]. When compared to SARS-CoV, which shares greater than 90% amino acid identity with SARS-CoV-2, proteins antagonizing IFN, such as nsp3, ORF3b, ORF6, have relatively low-sequence homology [155]. In SARS-CoV-2, ORF3b contains a premature stop codon that results in a truncated protein, and ORF6b is missing two amino acids at the C-terminal critical for the protein function [107]. This may explain the enhanced susceptibility of SARS-CoV-2 to IFNs as compared to SARS-CoV [155]. This runs parallel to in vitro observations that reported that SARS-CoV infection does not induce significant IFN-I production and with clinical studies which reported a lack of IFN response in SARS patients despite the robust production of cytokine and chemokine [108,156].

In this regard, it is crucial to mention that the cytokine and chemokine profile differs in the different stages during COVID-19 infection, as reported by several studies [145]. In turn, the temporal kinetics of the cytokine/chemokine profile can reflect the inflammatory mediators contributing to disease severity and provide an idea about the treatment modality to be used at different stages. This applies as well to IFN-1 induction, where the timing of exogenous IFN administration relative to viral replication is a key determinant of the response outcome. Accordingly, IFN-1 supplementation is most beneficial when given early in the disease course, when IFN-I expression is delayed or reduced due to viral suppression of IFN response or the older age of the host [145].

### 6.3. IL-17

Severely infected COVID-19 patients exhibit significantly elevated numbers of CCR6 + Th17 cells [157]. The immune response of Th17 is influenced by the cytokines and chemokines released in response to SARS-CoV-2 infection, including IL-1β and TNF-α [158]. In turn, Th17 produces IL-17, GM-CSF, IL-21, and IL-22, among which IL-17 was detected to be involved in the immune response against SARS-CoV-2 via releasing proinflammatory cytokines, such as G-CSF, which induces granulopoiesis and recruits neutrophils; and IL-1β, IL-6, and TNFα, which collectively cause systemic inflammatory response [158]. Furthermore, Th17 induces the release of CXCL1, CXCL2, IL-8, CXCL10, and CCL20, which attract more immune cells to the injured lung, in addition to the release of metalloproteinases responsible for tissue damage and remodeling [158]. These effects of Th17 reflect its contribution to the cytokine storm and pulmonary edema, making it an important target for therapy in COVID-19 patients. However, in contrast to IL-6, IL-17 did not demonstrate a significant difference between severely and mildly infected patients [140].

### 6.4. TGF-β

TGF-β is an anti-inflammatory cytokine released upon SARS-CoV-2 infection [130]. In response to SARS-CoV-2 infection, TGF-β is released from different sources, including the dysregulated coagulation and fibrinolytic pathways; the neutrophils massively infiltrating the lungs; and the macrophages migrating to the lungs to phagocytize apoptotic bronchial epithelial cells, pneumocytes, T-lymphocytes, and neutrophils [159]. The effect of this cytokine in SARS-CoV-2 infection is explained by its ability to recruit more neutrophils and remodel the airways by regulating processes used by the virus to develop pulmonary fibrosis. Pulmonary fibrosis seen in the lung biopsy of patients occurs through promoting myelofibroblast differentiation and fibroblast proliferation [157,159,160,161]. Eventually, this leads to failure in pulmonary function and death, hence making TGF-β an important target for therapy.

### 6.5. TNF-α

TNF-α is a proinflammatory cytokine released by macrophages and monocytes during acute inflammation and was found to be elevated in COVID-19 patients in general; however, more pronounced levels were detected in severe cases [29]. TNF-α amplifies inflammation by enhancing oxidative stress and leukocyte adhesion to the epithelium, modulating blood coagulation, and inducing fever indirectly [162]. Based on the structural homology between SARS-CoV and SARS-CoV-2, it is hypothesized that the same strategy to release TNF-α is used by both viruses, characterized by the modulation of TNF-α-converting enzyme (TACE)-dependent shedding of the ACE2 ectodomain, using viral spike protein [163].

### 6.6. CXCL10

Similar to SARS and MERS, CXCL10 is significantly elevated in COVID-19 patients. Analysis of 48 cytokines in the plasma of 50 COVID-19 patients showed that CXCL10 is an important biomarker for disease severity, and its increase was associated with an elevation in CCL7 (monocyte-chemotactic protein 3 (MCP3)) [164]. Taken together, CXCL10 and CCL7 can be excellent predictors for disease progression, especially since the CXCL10–CXCR3 signaling pathway was shown to play an important role in ARDS pathogenesis [165].

## 7. How Does the Immune Profiling in COVID-19 Correlate with SARS and MERS?

Based on the comparison between various inflammatory mediators involved in SARS, MERS, and COVID-19 immunopathology, it is concluded that common chemokines and cytokines profiles are shared among patients infected with the three viruses. Moreover, the same important chemokines and cytokines contributing to disease severity are involved in the three β-CoVs, such as the delay in the production of antiviral IFN and the elevation of IL-6, IL-17, CXCL10, TNF-α, and TGF-β. This implies that SARS-CoV, MERS-CoV, and SARS-CoV-2 use similar mechanisms for escaping the immune system, virus replication, dissemination, and disturbing the innate immune response, causing the cytokine storm. This can be attributed to the genetic and structural homology between the three β-CoVs. However, despite the similarities in the cytokine and chemokine profile, SARS, MERS, and COVID-19 have different rates of transmission and mortality, reflecting evolutionary differences in viral behavior that need to be further investigated. In addition, SARS-CoV-2 cytokine storm is not restricted to the elderly with co-morbidities but is also a threat to young healthy individuals with no risk factors. This can be attributed to the different mutations of SARS-CoV-2 and the different genetic makeup in individuals that define their tolerance for the infection [166].

Furthermore, based on the role of cytokines and chemokines in the immunopathology of COVID-19, the need to measure inflammatory mediators by developing a scoring system with cutoff values became a necessity, especially considering the practicality of blood tests in comparison to CT scans. IL-6, IFN, and CXCL10 stand to be important markers for disease severity. In addition, ratios of some cytokines and chemokines can be used, such as IL-6/IFN-γ ratio, which was demonstrated in a meta-analysis to be elevated in severe COVID-19 patients compared to mild cases and linked to the interaction between IFN-γ and IL-6/sIL-6R signaling [167,168]. However, practically not all cytokines can be easily assessed in peripheral blood, such as IFN-γ and IL-1β [143]. However, evaluating the inflammatory mediators that can be measured in blood or serum especially those of prominent importance in disease severity will help in detecting patients at risk of progressing to a severe state and allow for early preventive measures that can eventually reduce the mortality rate.

## 8. Treatments Targeting Chemokines and Cytokines in COVID-19

The immune system plays a major role in infections caused by β-CoVs and the various players, including immune cells, chemokines, cytokines, and immune checkpoints, can be considered potential targets for therapy. However, based on our review, cytokines and chemokines are the main underlying reason for complications and mortality. Drugs in clinical trials or those currently used can target these inflammatory mediators directly or indirectly either by targeting the immune cells that produce them or simply by inhibiting the virus entry and replication, which hinder the inflammatory reaction. This section discusses the different drugs that are currently being tested or used to directly inhibit the proinflammatory effects of chemokines and cytokines based on the immune profile of COVID-19 patients, while focusing on the inflammatory mediators with prominent importance in disease severity (Table 1). This will shed light on the possible treatments that might help in preventing the transition from a mild disease state to a severe one, which, in most cases, leads to death. Several drugs currently used against COVID-19 patients or being tested in clinical trials that target the various chemokines and cytokines responsible for the cytokine storm in COVID-19, including IL-6 inhibitors, IFN, IL-1β inhibitors, JAK inhibitors, and IL-17 antibodies, are illustrated (Figure 1).

As demonstrated above, IL-6 is a major cytokine involved in the cytokine storm in COVID-19 patients. Tocilizumab, sarilumab, and siltuximab are different IL-6 antagonists with different pharmacologic properties used efficiently and selectively in the clinic against autoimmune diseases, such as rheumatoid arthritis (RA) and inflammatory conditions, e.g., cytokine-release syndrome [187,188,189]. The effectiveness of tocilizumab in severe cases of COVID-19 with multi-organ failure was examined in a clinical trial in China. Tocilizumab was found to attenuate the cytokine storm and improve patients’ symptoms, such as hypoxemia, lymphopenia, fever, and lung infiltration, without adverse side effects [181]. One of the mechanisms of action of tocilizumab is its ability to restore the expression of HLA-DR on monocytes in COVID-19 patients and reverse the lymphocytopenia as fast as 24 h of treatment [142]. Another option for treating SARS-CoV-2 infection is antagonists against IL-6 receptors, which can be more effective than IL-6 inhibitors. IL-6 inhibitors can only suppress the *cis*- and *trans*-signaling pathways, while IL-6R inhibitors can also suppress *trans* presentation specific for Th17 cells as an important immune cell contributing to ARDS and helping in ameliorating the cytokine storm in COVID-19 patients [138,190,191]. However, the protective effect of IL-6 antagonists can be argued due to their ability to induce a rapid reduction in IL-10, which is an immunosuppressive cytokine secreted by macrophages, leading to a delay in viral clearance [138,192]. This discrepancy in the effect of IL-6 requires further investigation.

Moreover, the current limitations of IL-6 antagonists are the risk of infections and possible side effects in light of undetermined drug dosage and timing. Therefore, tocilizumab is recommended for use in severely infected patients with high viral load, and not in early disease stages, where it might adversely compromise the effect of IL-6 in viral clearance [182,193]. Another challenge is the compensatory role of other cytokines, such as IL-18, IFN-γ, and the JAK1 pathway, to regulate macrophage function. Thereby, inhibition of IL-6 alone might not be sufficient to improve patients’ conditions, especially since a recent study showed that IL-18 and IFN-γ are also among the cytokines elevated in severe COVID-19 patients [194], thus raising the need for combination therapies, which target more than one cytokine at a time. In this regard, more randomized control studies are needed to specify the timing and dosage of IL-6 antagonists, as well as possible drug combinations to reach the best outcome with the lowest side effects.

Aside from IL-6 and IL-6R antagonists, myo-Inositol, a supplement for hormonal regulation that is known to specifically downregulate IL-6 levels and PI3K, which is a key player in the IL-6 signaling pathway, was validated to attenuate the inflammatory pathway in lung disease [195,196,197,198,199]. Because IL-6 is upregulated by hypomethylation in its promoter region, preliminary data indicate that myo-inositol might exert epigenetic effects in targeting this cytokine, hence suggesting that myo-inositol might be beneficial for COVID-19 patients with high IL-6 levels, particularly because it has no adverse side effects compared to IL-6 antagonists [183,200,201].

### 8.1. IFNs

Dysregulation in IFN activity is a common mechanism used by the three β-CoVs to evade the immune system and cause an uninterrupted increase in viral replication. For this reason, exogenous use of IFN is one treatment option that was tested in both SARS and MERS and recently in COVID-19. In vitro studies showed that IFN-β and IFN-γ reduced SARS-CoV replication and plaque formation [169,170]. Furthermore, a retrospective clinical study and a four-arm trial demonstrated that a combination of IFN alfacon-1 and steroids improved patients’ outcomes by reducing lung abnormalities, restoring oxygen saturation level, and normalizing creatinine phosphokinase and lactate dehydrogenase levels [171,172]. However, administration of IFN alfacon-1 and steroids at a late stage of infection proved ineffective, hence stressing the relevance of their combined administration during an early stage of infection [171]. Other retrospective studies showed that interferon alfacon-1 could also be useful if combined with protease inhibitors, together with the viral replication inhibitor, ribavirin, or convalescent plasma-containing neutralizing antibodies [173]. Recombinant IFNα/β and IFN agonists, alone or in combination with antiviral drugs, were also shown to be effective against MERS-CoV. PEGylated-IFN-α (PEG-IFN-α) decreases viral RNA and cytopathic effect before, during, or after the infection of bronchial epithelial cells with SARS-CoV and MERS-CoV; however, in vitro studies showed that MERS-CoV is 50–100 times more sensitive to IFN-α than SARS-CoV, due to the absence of ORF6 protein, which blocks IFN-induced nuclear translocation of p-STAT1 [174]. Further, the combination of IFN-α with ribavirin reduced MERS-CoV replication in epithelial cell lines and infected rhesus macaques [175,176]. However, two clinical studies reported contradictory results upon testing the combination effect of oral ribavirin and subcutaneous PEG-IFN-α2a in MERS-CoV infected patients [202,203]. On the other hand, IFN-β was reported to induce stronger inhibition of MERS-CoV in vitro compared to IFN-α [177]. In addition, animals treated with lopinavir/ritonavir and IFN-β1b exhibited reduced viral load and improvement in the clinical outcomes [178]. Poly(I:C), a strong type I IFN agonist exerting its activity through TLR3 activation was also found to reduce MERS-CoV load and enhance its clearance in BALB/c mice transfected with adenoviral vectors expressing human DPP4 (Ad5-hDPP4) [66]. Moreover, mycophenolic acid, an immunosuppressant commonly used in recipients of organ transplantation, was found to be effective against MERS-CoV in vitro through modulating the expression of ISGs [177,179,180].

Based on the antiviral effects of IFN-λ discussed earlier and based on the illustrated research on the use of exogenous IFN in SARS and MERS and on some FDA-approved IFN-γ drugs (e.g., Emapalumab) used to treat hemophagocytic lymphohistiocytosis (HLH) and MAS, IFNs are being adopted as an antiviral treatment in COVID-19 patients [131,143]. Due to IFNs’ protective role, initially, when the viral load is still low, IFNs can be considered as a prophylactic treatment for patients in the early stage of infection, with no signs of inflammatory reaction in the lungs; or in high-risk patients with co-morbidities, being at an increased risk to develop complications [184,204]. Particularly in SARS-CoV-2 infection, the time between first the disease symptoms and ARDS is quite sufficient to allow for this type of intervention as a preventive measure [153].

In comparison with type I IFN, type III IFN-λ is more effective and potent to restrict viral replication in the upper respiratory tract, since the receptors for type III IFN-λ (IFNLR), in contrast to type I IFN receptors (IFNAR), are not present on immune cells, thus reducing the possible systemic side effects, including inflammation and tissue damage [184,204,205,206,207]. The use of IFN-λ in SARS-CoV-2 infection is encouraging, since IFN-λ enhances the adaptive immune system by stimulating Th1, cytotoxic T cells, and antibody responses required for developing long-term immunity [208,209]. Moreover, it was shown that IFN-λ is effective on other β-CoVs, such as MERS-CoV and SARS-CoV [184]. The only available IFN-λ therapeutic agent is PEG-IFN-λ1, and it was shown to be safe in lung infections based on 19 clinical studies [184]. Although the use of IFN-λ can be promising in COVID-19 patients, extensive research is still needed to investigate several gaps in the biologics of IFNs, especially in the context of SARS-CoV-2. Before trying to use IFN-λ as a treatment option, it is important to investigate the following: whether the SARS-CoV-2 induces or blocks the expression of IFN-λ, whether IFNLR1 is present on alveolar macrophages and endothelial cells, and whether the expression of both IFN-λ and its receptors varies with age [184]. Moreover, exogenous IFN-λ can be accompanied by adverse events due to the upregulation of IFNLR in the inflamed environment, which exacerbate the inflammatory process, thus raising the need to investigate the expression of IFNLR upon IFN-λ treatment and the subsequent responsiveness of immune cells to this stimulation. [210,211]. In contrast to type I IFN, IFN-λ facilitates bacterial superinfection by reducing the recruitment of neutrophils and their bactericidal activities, putting this treatment at a disadvantage [212,213,214].

### 8.2. Th17 Blockades

Several strategies can be used to inhibit Th17 effects in response to SARS-CoV-2, and these include antibodies targeting the cytokines released by Th17, such as the available anti-IL-17, anti-IL-17R, and anti-IL-12/23p40, or antibodies targeting the transcription factors ROR-γt and ROR-α, which are currently being tested in clinical trials. However, the narrow spectrum and high cost of antibody-based therapy make JAK inhibitors and specifically JAK2 inhibitors that block Th17 a better option to consider for several reasons [158]. First, it is crucial to highlight that the JAK2–STAT3 signaling pathway is downstream to IL-6 and IL-23 cytokines required for Th17 differentiation and function and is also downstream to IL-21, which signals in B-cells, using JAK1 and JAK3 instead of JAK2 [215]. Accordingly, one of the advantages of JAK2 inhibitors, such as Fedratinib, is the selective targeting of IL-6 and IL-23 without compromising the effect of IL-21, as in the case when using STAT3 inhibitors [158]. In the same context, the selectivity of Fedratinib rescues type I IFN, which employs JAK1–STAT1/2 and maintains its antiviral effect [158]. In comparison with IL-6 antagonists whose early administration might compromise the IL-6 effect in viral clearance, the reversibility of JAK2 inhibitors makes using this treatment as a preventive measure possible even before the disease progresses to a severe state and without affecting the needed Th17 immune response in fighting the virus [158]. Another advantage of Fedratinib is that it inhibits the effect of IL-6 on other cell types and the feasibility of using it in combination with other antiviral drugs [158]. Hence, the ability of Fedratinib to target several cytokines with or without other drug combinations pinpoints its promising effects in treating COVID-19 patients.

### 8.3. Janus Kinase (JAK) Inhibitors: Baricitinib, Tofacitinib, and Ruxolitinib

Similar to tocilizumab, JAK inhibitors are approved to treat RA and other inflammatory disorders [216]. JAK inhibitors interfere with the JAK–STAT signaling pathway, which mediates the effect of various cytokines, including IL-2, IL-3, IL-4, IL-5, IL-6, IL-7, IL-9, IL-10, IL-12, IL-15, IL-21, IL-23, and IFN-(α, β, and γ) [217]. The effect of JAK inhibitors on the cytokine storm can be either direct via inhibiting members of the JAK family enzymes (JAK1, JAK2, JAK3, and TYK2) and their downstream signaling or by blocking kinases that regulate endocytosis of SARS-CoV-2 virus, such as AP-2-associated protein kinase 1 (AAK1) and cyclin G-associated kinase (GAK) [218,219]. Ongoing clinical trials to evaluate the effectiveness and efficacy of some JAK inhibitors, such as baricitinib, tofacitinib, and ruxolitinib, are being conducted [219].

### 8.4. IL-1β Inhibitors: Anakinra and Colchicine

As mentioned above, inflammasomes and the pyroptosis process are involved in host immune response against SARS-CoV-2, leading to the release of proinflammatory cytokines, mainly IL-1β and IL-18, which are involved in hematopoiesis and fibrosis. Since IL-1β contributes significantly to the cytokine storm and vascular permeability, and then inhibiting it by anakinra, the approved recombinant antagonist of human IL-1 for RA can be also considered as a treatment regimen in COVID-19 patients [219,220,221]. A phase-three clinical trial on the effect of anakinra on sepsis and macrophage-activating syndrome (MAS) supports this option to attenuate the cytokine storm post-SARS-CoV-2 infection, especially that patients improved significantly without adverse events [222,223]. Another drug that is known to inhibit the NLRP3 inflammasome is colchicine, which also has a direct effect on TNF-α and IL-6 synthesis and, accordingly, stands to possibly have desirable effects in treating COVID-19 patients [224]. Similar to IL-6 antagonists, the dosage and timing of anakinra and colchicine are yet to be determined by clinical trials to attain optimal effects. Immunosuppressive cytokines IL-37 and IL-38 are also known to inhibit IL-1β, IL-6, TNF, and CCL2 by binding to IL-18Rα receptors; targeting mTOR; increasing adenosine monophosphate kinase; and inhibiting MHC class II molecule by suppressing MyD88 [225]. These anti-inflammatory cytokines can be potential treatment options for COVID-19.

### 8.5. Other Treatments

Among other cytokine/chemokine-targeting drugs used or in clinical trials to treat COVID-19 is the adjunctive therapy cytosorb, which reduces the circulating levels of cytokines, DAMPs, and PAMPs by absorbing them to treat the cytokine storm [226]. GM-CSF, a key molecule in the cytokine storm of COVID-19 patients, can be blocked either by Mavrilimumuab or by JAK inhibitors, since GM-CSF uses JAKs in its signaling pathway [136,158]. Further, SAR-CoV-2 infection is associated with the release of reactive oxygen species which activate the NF-κB and activator protein-1 pathways and induce, as a consequence, the expression of proinflammatory cytokines, such as IL-6, IL-8, and TNF-α [227]. N-Acetylcysteine (NAC) is a free-radical scavenger that can aid in inhibiting these cytokines and improving oxidative stress caused by to SARS-CoV-2 infection [228]. Moreover, given the importance of TNF-α in mediating inflammatory reactions by promoting the release of other chemokines and cytokines—and since the levels of TNF-α are shown to be elevated in COVID-19 patients, particularly in severe cases—clinical trials on anti-TNF-α drugs, such as adalimumab and infliximab, are being conducted as a potential option for treatment [185,186]. Lastly, recent news from an ongoing clinical trial suggested a breakthrough in COVID-19 management through the use of dexamethasone. Dexamethasone inhibits IL-1 and TNF activity in the lung fibroblasts and thereby reduces lung fibrosis, a common COVID-19 complication [229]. To further elucidate the mechanism of action of dexamethasone in treating SARS-CoV-2, a study revealed that, by using a single-cell atlas, single RNA sequencing, and plasma proteomics, dexamethasone affected circulating neutrophils, altered IFN^active^ neutrophils, downregulated interferon-stimulated genes, and activated IL-1R2^+^ neutrophils [230]. This will help in controlling COVID-19-associated ARDS characterized by the expansion of distinct neutrophil states characterized by IFN and prostaglandin signaling. Interestingly, males having a higher proportion of IFN^active^ neutrophils can benefit from the steroid-induced immature neutrophil expansion [230]. This study highlights the effect of gender difference on SARS-CoV-2 infection, outcome, and treatment choice.

## 9. Conclusions

SARS-CoV-2 shares structural features and viral behavior with SARS-CoV and MERS-CoV, yet SARS-CoV-2 has a higher transmission rate and more virulence properties as compared to other β-CoVs. The lack of knowledge regarding SARS-CoV-2 virology and host immune responses promoted an urgent need for prophylactic treatment options to manage infected individuals, especially those at risk of developing severe complications. Chemokines and cytokines play a major role in COVID-19 immunopathology, as they are the underlying cause for exacerbated immune response, leading to cytokine storm, ARDS, multiple-organ failure, and eventually death.

This extensive and up-to-date literature review regarding SARS-CoV-2 immune signature identified several limitations, including variability in laboratory assays used to evaluate cytokine levels, lack of negative control recruitment due to the restrictions imposed by the pandemic, differences in populations demographics, associated co-morbidities or co-infections, and limited study scope to gain a holistic immune profile. Despite these limitations, there is a clear consensus regarding the chemokine and cytokine profile among COVID-19 patients in which IL-6, IFN, CXCL10, IL-17, TGF-β, and TNF-α are major contributors to the pulmonary immunopathology post-SARS-CoV-2 infection and stand to be important targets for therapy. The importance of inflammatory mediators’ crosstalk should also be considered, especially when using drug combinations, in order not to compromise beneficial host immune responses. Furthermore, the host immune profile should serve as the basis for optimum disease management and drug design.

Fine-tuning of cytokines’ and chemokines’ release is crucial to clear the infection without progressing to organ dysfunctions and death. Based on reviewing the treatment modalities used to manage SARS, MERS, and COVID-19, this review recommends the use of immunosuppression cocktails, provided that patients are closely monitored and continuously assessed to maintain the desirable effects of cytokines and chemokines needed to fight the virus. Finally, more clinical trials are needed to determine the optimal and effective dosage and timing for a therapeutic regimen, as this is lacking for most clinically used drugs.

## Figures and Tables

**Figure 1 viruses-14-00164-f001:**
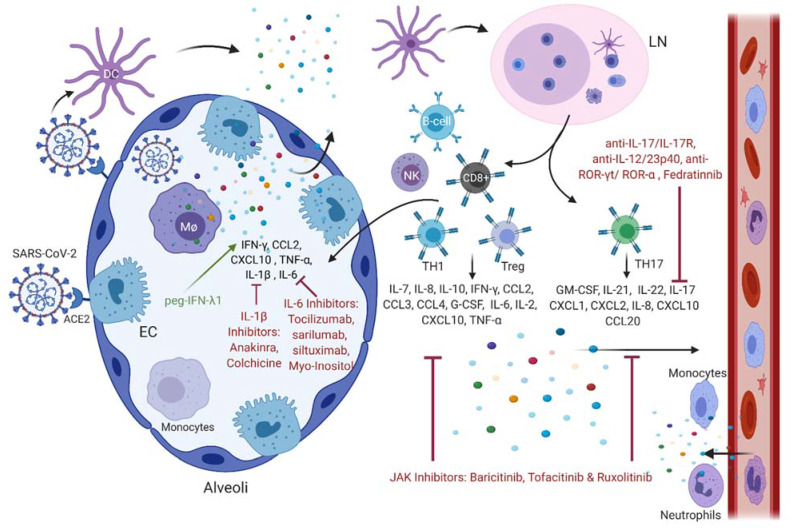
Innate and adaptive immune cells involved in SARS-CoV-2 infection, as well as the different chemokines and cytokines released and their inhibitors. SARS-CoV-2 binds ACE2 receptor on epithelial cells (ECs) and on macrophages (MΦ) of the alveoli. This triggers the release of cytokines and chemokines that attract more immune cells to the injured lung. Upon infection, activated DCs migrate to the lymph node (LN), where they activate T- and B-lymphocytes, which will further release pro-inflammatory mediators that exacerbate the infection. Several drugs currently used on patients or being tested in clinical trials target the various chemokines and cytokines responsible for the cytokine storm in COVID-19, including IL-6 inhibitors, INF, IL-1β inhibitors, JAK inhibitors, and IL-17 antibodies.

**Table 1 viruses-14-00164-t001:** Summary of the common cytokine and chemokine profile of patients infected with SARS-CoV, MERS-CoV, and SARS-CoV-2.

Common Chemokine Profile	Common Cytokine Profile	Important Chemokine/Cytokine Involved in Disease Pathogenesis (Particularly in Severely Infected Cases)	Role of the Most Prominent Chemokine/Cytokine in Disease Pathogenesis	Drugs Targeting Chemokines and Cytokines
**SARS-CoV**				
CCL2, CCL3, CCL5, CCL10, CXCL-8 (IL-8), CXCL9, and CXCL10	IL-1β, IL-2, IL-6, IL-10, IL-12, TNF-α, IFN-α/α2, IFN-β1, IFN2, IFN-γ, and TGFβ	Dysregulation of IFN (α and γ)	Induces excessive cytokine and chemokine levels [109].	IFN alfacon-1 and steroids at early stage infection [169,170,171,172,173].
		Elevation in CXCL10	Recruits monocytes, macrophages, dendritic cells, NK cells, and T-lymphocytes toward interstitial lung tissue [112].	
**MERS-CoV**				
CCL-2, CCL-3, CCL-5, CXCL-8 (IL-8), and CXCL-10	IL-1β, IL-6, IL-10, IL-12, IL-13, IL-15, IL-17, IL-23, TNF-α, IFN-γ, IFN- α2, and TGFβ	Dysregulation of IFN (α, β γ)	Develops early antiviral Th-1 [121].	Recombinant IFNα/β and IFN agonists (e.g., poly(I:C)) and mycophenolic acid [66,174,175,176,177,178,179,180].
		Elevation in IL-10	Inhibits IFN-γ production [121,125]. Reduces CD8+ T-cells proliferation. [121,125] Increases viral replication [121,125].	
		Elevation in IL-6		
		Elevation in CXCL10		
		Elevation in IL-17	Recruits neutrophils and monocytes [33,123]. Contributes to the release of IL-1β, IL-6, TNF-α, TGF-β, IL-8, and CCL2 [33,123].	
**SARS-CoV-2**				
CCL2, CCL3, CCL4, CCL7, CCL8, CXCL1, CXCL2, CXCL-8 (IL-8), CXCL6, CCL20, CXCL-10, and CXCL17	IL-1β, IL-2, IL-6, IL-7, IL-10, IL-17, IL-33, IFN-ɣ, TNF-α, and TGFβ	Elevation in IL-6	Contributes to the release of VEGF, CCL2, IL-8 and additional IL-6 [138]. Decreases E-cadherin expression on endothelial cells, leading, together with VEGF, to an increase in vascular permeability and leakage, hypotension, and pulmonary dysfunction [138]. Inhibits HLA-DR expression on CD14 monocytes, leading to defective lymphoid function [141]. Impairs the cytotoxic function of NK cells [143,144]. Increases CRP, serum amyloid A, fibrinogen, and hepcidin and inhibits albumin synthesis [138].	IL-6 and IL-6R inhibitors: tocilizumab, sarilumab, siltuximab, and myo-inositol [142,181,182,183]. JAK Inhibitors [158].
		Upregulation of IFN signaling pathway, but downregulation of IFN levels	Increases ISGs and IFITMs, which inhibit the cellular entry of the virus [131,149,150].	PEGylated IFN-λ1(PEG-IFN-λ1) [184].
		Elevation in IL-17	Induces the release of G-CSF responsible for granulopoiesis and neutrophils recruitment [158]. Contributes to pulmonary edema by inducing the release of metalloproteinases responsible for tissue damage and remodeling [158]. Induces IL-1β, IL-6, and TNFα, which collectively cause systemic inflammatory response [158]. Induces the release of CXCL1, CXCL2, IL-8, CXCL10, and CCL20, which recruit more immune cells to injured lung [158].	Anti-IL-17, anti-IL-17R and anti-IL-12/23p40 [158]. Anti-ROR-γt and ROR-α [158]. JAK Inhibitors: Fedratinib [158].
		Elevation in TNF-α	Amplifies inflammation by enhancing oxidative stress and leukocyte adhesion to the epithelium, modulating blood coagulation and inducing fever indirectly [162].	Anti-TNF-α drugs: Adalimumab and infliximab [185,186].
		Elevation in TGF-β	Recruits neutrophils and remodel the airways by regulating processes used by the virus to develop pulmonary fibrosis through promoting myelofibroblast differentiation and proliferation [157,159,160,161].	Anti-active TGF-β antibodies and/or TGF-β inhibitors [159].

## Data Availability

Not applicable.

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
