# Peer review of "Immune Profiling of COVID-19 in Correlation with SARS and MERS"

_viruses, 2022, doi:10.3390/v14010164_

Round 1
Reviewer 1 Report
This narrative review would benefit from adding Search Strategy through MEDLINE/PubMed, LitCovid and Scopus. The following recommendations could be consulted: https://pubmed.ncbi.nlm.nih.gov/21800117/
Please highlight what is new in your review.
Author Response
On behalf of the authors, I would like to thank you for allowing us to revise our manuscript. You raised a couple of important comments. Please see attached point-by-point responses and find attached our manuscript for your consideration.
Reviewer 2 Report
Summary:
This is a well-written and comprehensive review on an important topic. The introduction is clear and lays an appropriate foundation for the comparison of SARS, MERS and SARS-CoV-2 cytokine and chemokine profiles. This review summarizes literature on major cytokines and chemokines that may contribute to severe disease caused by three pathogenic coronaviruses: SARS, MERS, and SARS-CoV-2, and then discusses treatments aimed at reducing pathogenic cytokine/chemokine responses.
Major Concerns:
- The review does not discuss the data in terms of timing of disease course. Now that we know so much about COVID-19, we know that for many with severe disease, dysregulated cytokine production can follow two phases: reduced IFN/innate immune responses early in infection (that permit viral burst) followed by overactive cytokine/chemokine storms later during infection that leads to severe disease. Please consider the temporal dynamics of cytokine/chemokine levels in your review. See Park and Iwasaki, 2020, Cell Host Microbe for a great overview.
- The authors do not discuss any studies that stratify data by age or sex of the patient. Some studies have investigated if cytokine responses differ between male and female sex or between older and younger patients – please cite these data. Moreover, the authors fail to discuss an important recent paper that performed single-cell RNA sequencing on COVID patients to look at immune responses (stratifying data by sex) and then examined how use of dexamethasone treatment modulated these responses. Please consider a discussion of Sinha et al. Dexamethasone modulates immature neutrophils and interferon programming in severe COVID-19. Nature Medicine. 2021 which would fit nicely in the last section of the review.
3.
p.8
IFN production is diminished by several different SARS-CoV-2 proteins (e.g. M, N, Orf3b, nsp1, nsp7/8, nsp16…). There is a wealth of literature on this, and I do not feel like this section of the review is comprehensive. In addition, the authors diminish the importance of IFN response delay, as noted above, temporal dynamics are key. Several papers have also shown that defective or delayed IFN responses may contribute to severe disease.
Minor Concerns:
Line 88 – ‘highlight’ not highlighted
Lines 90-93 – sentence fragment
Lines 108-110
"Moreover, MERS-CoV exhibits the furin-like cleavage site also present in SARSCoV-2 but not in SARS-CoV; hence rendering the infection with SARS-CoV-2 more severe compared to SARS-CoV [53]."
Not sure about this statement. Is infection with SARS-CoV-2 more severe than SARS-CoV? Or is it more infectious because of the furin cleavage site? Reword.
Lines 485-7 and 490-2
"Antagonists against IL-6 receptors are another option for treating SARS-CoV-2 infection and can be more effective since unlike IL-6 inhibitors, which can only suppress the cis and trans signaling pathways.However, the protective effect of IL-6 antagonists can be argued due to their ability to induce a rapid reduction in IL-10, an immunosuppressive cytokine secreted by macrophages, consequently leading to a delay in viral clearance [135,181]."
Sentences are grammatically incorrect.
Figure 1:
The figure needs to be of higher resolution – it appears pixelated in the pdf of the manuscript. The figure legend is inadequate – it does not describe the multiple aspects of the figure.
Author Response
On behalf of the authors, I would like to thank you for allowing us to revise our manuscript. You raised some important comments. Please see attached point-by-point responses and find attached our manuscript for your consideration.
